# Thermodynamic and Kinetic Characterization of Colloidal Polymers of *N*-Isopropylacrylamide and Alkyl Acrylic Acids for Optical pH Sensing

**DOI:** 10.3390/molecules30071416

**Published:** 2025-03-22

**Authors:** James T. Moulton, David Bruce, Richard A. Bunce, Mariya Kim, Leah Oxenford Snyder, W. Rudolf Seitz, Barry K. Lavine

**Affiliations:** 1Department of Chemistry, Oklahoma State University, Stillwater, OK 74078, USA; jtmoult@okstate.edu (J.T.M.); dabruce@unmc.edu (D.B.); rab@okstate.edu (R.A.B.); mashakim2003@yahoo.com (M.K.); leah.snyder@fortsillapache-nsn.gov (L.O.S.); 2Department of Chemistry, University of New Hampshire, Durham, NH 03824, USA; rudi.seitz@unh.edu

**Keywords:** optical pH sensing, pH-induced polymer swelling, *N*-isopropylacrylamide, nonionic polymer swelling

## Abstract

Copolymers of *N*-isopropylacrylamide (NIPA) and alkyl acrylic acids that swell and shrink in response to pH were prepared by dispersion polymerization at 35 °C using *N*-isopropylacrylamide (transduction monomer), methylenebisacrylamide (crosslinker), 2-dimethoxy-2-phenyl-acetophenone (initiator), *N*-*tert*-butylacrylamide (transition temperature modifier), and acrylic acid, methacrylic acid, ethacrylic acid, and propacrylic acid (functional comonomer). The diameter of the microspheres of the copolymer varied between 0.5 µm and 1.0 µm. These microspheres were cast into hydrogel membranes prepared by mixing the pH-sensitive swellable polymer particles with aqueous polyvinyl alcohol solutions followed by crosslinking the polyvinyl alcohol with glutaric dialdehyde for use as pH sensors. Large changes in the turbidity of the polyvinyl alcohol membrane monitored using a Cary 6000 UV–visible absorbance spectrometer were observed as the pH of the buffer solution in contact with the membrane was varied. Polymer swelling was reversible for many of these NIPA-based copolymers. The buffer capacity, ionic strength, pH, and temperature of the buffer solution in contact with the membrane were systematically varied to provide an in-depth pH profile of each copolymer. A unique aspect of this study was the investigation of the response of the NIPA-based polymers to changes in the pH of the solution in contact with the membrane at low buffer concentrations (0.5 mM). The response rate and the reversibility of polymer swelling even at low buffer capacity suggest that NIPA-based copolymers can be coupled to an optical fiber for pH sensing in the environment. We envision using these polymers to monitor rising acidity levels in the ocean due to water that has become enriched in carbon dioxide that endangers shell-building organisms by reducing the amount of carbonate available to them.

## 1. Introduction

Optical pH sensors implemented with fiber optics [1,2,3] for environmental and biomedical analyses problems outside of the laboratory, e.g., continuous in vivo pH measurements of blood in arteries and muscles [4], and monitoring and controlling pH in fermentation baths [5], have been investigated as an alternative to the glass electrode, which has proven to be problematic in these applications due to drift and interference by other components, thereby requiring the continued recalibration of the electrode. Several optical pH sensing methods based on absorbance or fluorescence that incorporate a variety of designs have been proposed using pH indicators (i.e., organic dyes) immobilized at the distal end of the optical fiber [6,7,8,9,10]. However, the use of these indicators for pH sensing has drawbacks including the loss of the indicator due to leaching over time and photodegradation. Furthermore, the indicators are wavelength-specific, restricting the instrumentation that can be used. To overcome these disadvantages, fiber optic pH sensors have been developed that utilize swellable polymers functionalized to respond to pH [11,12,13]. There are several advantages to pH sensing based on polymer swelling [14,15,16,17] compared to methods that utilize indicators. Since the analyte signal is based on turbidity, which is independent of wavelength, measurements can be made in either the long wavelength region of the visible using light-emitting diodes as the source and photodiodes as the detector for an optical reflecting device or in the near infrared in which the wavelengths can be transmitted through an optical fiber without appreciable attenuation, unlike the ultraviolet and visible electromagnetic radiation used in absorption or fluorescence. This allows for inexpensive components and a fiber optic sensing system that is inexpensive compared to laser-based fiber optic systems. The use of swellable pH-sensitive polymer particles entrapped in a permeable hydrogel membrane for a turbidity measurement avoids the problem of photodegradation that can occur when absorption or fluorescence is used for sensing. These membranes have been shown to be stable for more than ten years. The polymer microspheres do not leach out of the membrane compared to reagent phases such as indicators, and the swellable polymer microspheres are protected from direct contact with the sample by the hydrogel, which can serve as a “filter” to reject sample components such as humic acid or suspended particles in the sample that are too large to diffuse through the hydrogel.

In this study, copolymers of *N*-isopropylacrylamide (NIPA) and alkyl acrylic acids that swell and shrink in response to pH were prepared by dispersion polymerization at 35 °C using *N*-isopropylacrylamide (transduction monomer), methylenebisacrylamide (crosslinker), 2-dimethoxy-2-phenyl-acetophenone (initiator), *N*-*tert*-butylacrylamide (transition temperature modifier), and acrylic acid, methacrylic acid, ethacrylic acid and propacrylic acid (functional comonomer). pH-induced swelling [18] of these poly(*N*-isopropylacrylamide) copolymers is comparable in magnitude to the swelling of microgels of poly(*N*-isopropylacrylamide) that undergo a volume phase transition from a swollen state to a coiled (collapsed) state over a narrow temperature range when placed in an aqueous solution at physiological temperature, which makes these polymer particles (microspheres) viable in fiber optic pH sensors. The diameter of the microspheres of the copolymer varied between 0.5 μm and 1.0 μm. The microspheres of the copolymer were cast into hydrogel membranes prepared by mixing the pH-sensitive swellable polymer particles (microspheres) with aqueous polyvinyl alcohol (PVA) solutions followed by crosslinking the PVA with glutaric dialdehyde for use as pH sensors. The microspheres swell and shrink with changing hydrogen ion concentration of the aqueous solution in contact with the hydrogel membrane. When the membrane is exposed to an alkaline solution, the microspheres swell due to a change in the polymer–solvent interaction parameter triggered by the deprotonation of the carboxylic acid group of the alkyl acrylic acid incorporated into the NIPA copolymer. This reduces the refractive index of the poly(*N*-isopropylacrylamide) copolymer so that it is closer to the refractive index of the hydrogel, leading to a decrease in the amount of light reflected and the turbidity because the fraction of light reflected at an interface increases with the difference in the refractive index of the two media.

Large changes in the turbidity of the PVA membrane monitored using a Cary 6000 UV–visible absorbance spectrometer were observed as the pH of the buffer solution in contact with the membrane was varied. Polymer swelling was also reversible for many of the NIPA-based copolymers prepared for use in this study. The buffer capacity, ionic strength, pH, and temperature of the buffer solution in contact with the membrane were systematically varied to provide an in-depth pH profile of each copolymer. Both the degree of swelling and the apparent pK_a_ of the polymer particles are observed to increase with temperature, and the degree of swelling and the apparent pK_a_ of the polymer particles are independent of the ionic strength of the solution in contact with the membrane when the polymer particles are sufficiently crosslinked. Furthermore, the apparent pK_a_ of the polymer particles can be tuned by varying the concentration of the components used in the formulation and the alkyl chain length of the pH-sensitive comonomer.

In our previous studies [19,20], the effects of temperature and ionic strength on the pH response of a series of NIPA copolymers containing different pH-sensitive functional comonomers were investigated at buffer concentrations of 50 millimolar or 100 millimolar. A unique aspect of this study is the investigation of the response of *N*-isopropylacrylamide-based polymers to changes in the pH of the solution in contact with the membrane at low buffer capacity (0.5 mM) as a function of the temperature and ionic strength of the buffer. Another unique aspect of this study is the investigation of the kinetics (or response rate) of the shrinking and swelling of *N*-isopropylacrylamide-based polymers, to changes in the pH of the buffer in contact with the membrane at different temperatures and ionic strengths. The response rate and the reversibility of polymer swelling, even at low buffer capacity, suggest that NIPA-based copolymers can be coupled to optical fibers for pH sensing in the environment. We envision many applications for these polymers such as monitoring the rising acidity levels in the ocean due to water that has become enriched in carbon dioxide, endangering shell-building organisms by reducing the amount of carbonate available in the water for them [21].

## 2. Results

### 2.1. Copolymer of N-Isopropylacrylamide and Methacrylic Acid

Figure 1 shows the pH response profiles (both ascending and descending pH) of the poly(*N*-isopropylacrylamide) synthate, M-70, prepared by copolymerization of *N*-isopropylacrylamide (NIPA) with methacrylic acid (MAA), *N*,*N*′-methylenebisacrylamide (MBA), and *N*-t-butylacrylamide (NTBA). The formulation used to prepare this swellable pH-sensitive polymer consisted of 15 millimoles of NIPA, 2 millimoles of MAA, 1 millimole of MBA, and 2 millimoles of NTBA. The polymer was embedded in a PVA hydrogel membrane. For each data point in Figure 1, the membrane was allowed to equilibrate for twenty minutes (to minimize the effects of kinetics) at ambient temperature (23.5 °C) prior to analysis by UV–visible absorbance spectroscopy. Turbidity (absorbance values) collected at 700 nm was plotted against the pH of each buffer solution. Three different buffer concentrations were employed: 50 millimolar, 5 millimolar, and 0.5 millimolar. The ionic strength of each buffer solution was fixed at 0.1 M using NaCl.

For the 50 millimolar and 5 millimolar buffer solutions, swelling is reversible as both the ascending (solid line) and descending (dashed line) pH profiles are superimposable, whereas polymer swelling is irreversible with the 0.5 millimolar buffer solutions. At low pH, M-70 exists in the shrunken state as the water content of the polymer is lower than that of the PVA hydrogel. Turbidity as measured by the Carey 6000 UV–visible spectrometer is higher because of the greater amount of light reflected due to the refractive index of the polyNIPA particles being greater than the refractive index of PVA. At higher pH values, the water content of the polymer particles increases due to swelling induced by the deprotonation of the pH-sensitive functional comonomer. The refractive index of the M-70 particles decreases as the particles swell, approaching the refractive index of PVA.

The inflection point (as determined by the second derivative) in each turbidity versus pH plot shown in Figure 1, Figure 2 and Figure 3 is the apparent pK_a_ of the polymer (see Table 1). The term “apparent pK_a_” refers to the inflection point in the pH profile where the response is midway between the response at lower pH and higher pH and is used here as the relationship between turbidity and pH has not yet been described by theory in a manner that allows for the calculation of pK_a_ from the observed titration data.

Figure 2 shows pH response profiles (both ascending and descending pH) for the 50 millimolar, 5 millimolar, and 0.5 millimolar buffers. It is evident from this plot that the irreversible swelling and shrinking of M-70 with the 0.5 millimolar buffer is linked to the descending pH profile (i.e., polymer shrinking). The fixed charges on the polymer (i.e., carboxylates) are surrounded by cations that stabilize these charges. The ionic strength of the buffers used for this comparison, including the 0.5 millimolar buffer, is adjusted to 0.1 molar using NaCl. However, the protons required for protonation are comparable to the amount that can be provided by the 1.9 mL of 0.5 millimolar buffer in the pH 5.7 to pH 4.7 range. This would explain the hysteresis effect for the 0.5 mM buffer as it is linked to protonation of the pH-sensitive sites on the NIPA copolymer by the 1.9 mL of the 0.5 millimolar buffer solution in the cuvette.

Figure 1 and Figure 3 show the pH response profiles (both ascending and descending pH) of M-70 for 50 millimolar, 5 millimolar, and 0.5 millimolar buffers at 23.5 °C, 31 °C, and 37 °C.

The ionic strength of each buffer solution was fixed at 0.1 M using NaCl. For the 50 millimolar and 5 millimolar buffers, polymer swelling for M-70 is reversible at all temperatures, whereas polymer swelling is irreversible at all temperatures with the 0.5 mM buffer solutions. A kinetic effect based on the concentration of acid and conjugate base in the 0.5 mM buffer explains the irreversibility observed for the 0.5 mM buffer. Alternatively, M-70 may be closer to equilibrium at higher temperature.

Table 2, Table 3 and Table 4 list the “apparent pK_a_” of M-70 for each buffer concentration at 23.5 °C, 31 °C, and 37 °C. The increase in the apparent pK_a_ of the polymer with temperature is the opposite to what occurs for an acid dissolved in water, which is a decrease of about 0.1–0.3 pH units for a 10 °C increase in temperature [22]. The increase in the apparent pK_a_ of M-70 occurs because there is less water in the polymer. The distance between adjacent MAA units, the source of charge in the three-dimensional polymer network, also decreases due to a decrease in the volume of the polymer due to the water molecules being driven out at higher temperature. The result is an increase in the apparent pK_a_ of the polymer.

Figure 4 shows M-70 pH response profiles (both ascending and descending pH) for 50 millimolar, 5 millimolar, and 0.5 millimolar buffers at 1.0 M ionic strength and 23.5 °C. Polymer swelling and shrinking of M-70 is irreversible in all three buffers at 1.0 M ionic strength as equilibrium has not been reached. Furthermore, M-70 does not return to its original state in any of the buffers at 1.0 M ionic strength, i.e., the turbidity of the membrane does not return to its original value after completing the reverse titration. The turbidity of M-70 at the start of the forward titration with the 1.0 M ionic strength buffer (see Figure 4) is greater than the turbidity of M-70 at the start of the forward titration with the 0.1 M ionic strength buffer (see Figure 1). It took nearly 12 h to soak the M-70 membrane in the low pH buffer solution after the reverse titration was over for the turbidity to return to its initial value. Evidently, salt is trapped in the polymer particles, and time is required for its release. Furthermore, the higher turbidity of the membrane at the beginning of the titration suggests that the polyNIPA chains are in a partially collapsed state. This would make the chains less accessible to the buffer. More time would be required to reach equilibrium in response to a change in the pH of the buffer solution in contact with the membrane. At high ionic strength, the polymer particles would also be smaller due to a decrease in their water content that arises from the differences between the osmotic pressure of the interior of M-70 and the buffer solution in contact with the membrane. Diffusion of the buffer into the particles would be slower.

#### Kinetic Studies of pH-Induced Polymer Swelling and Shrinking

Rate data for pH-induced polymer swelling and shrinking of M-70 are shown in Figure 5. The slower rate at higher ionic strength is unexpected as the effect of higher ionic strength is to reduce shielding and diminish the activity coefficients of the acid and conjugate base of the buffer. However, the rate data are consistent with the pH response curves of M-70 at 0.1 M and 1 M ionic strength (see Figure 1 and Figure 4). Evidently, more protons are required than can be provided by the 50 mM, 5 mM, or 0.5 mM buffer for protonation of the carboxylate ions as NaCl is added in substantial amounts to the buffer to fix the ionic strength at 1.0 M. In principle, this may be a result of an interplay between the number of protons in the solution and the rate of their transfer from the bulk solution to the membrane, although diffusion in solution, apparently is much faster than in the membrane. Differences in the rate of pH-induced polymer shrinking (see Figure 4) between low- and high-ionic-strength buffer solutions can probably be attributed to charge stabilization of the carboxylate anions by the sodium ions. For the shrinking study, the pH 3.3 buffer is exchanged for a pH 6.1 buffer. Prior to the initiation of the kinetic study, the membrane was allowed to equilibrate for 12 h in the pH 6.1 buffer. The pH step used in this experiment (pH 6.1 to pH 3.3) corresponds to the regions of the pH profile where M-70 transitions between the swollen state (pH 6.1) and the shrunken state (pH 3.3).

For pH-induced swelling (see Figure 6), the pH 6.1 buffer is exchanged for the pH 3.3 buffer. It takes longer to adjust the concentration of the cations at the interface between M-70 and the buffer solution at high ionic strength. Table 5 shows the apparent pK_a_ values of M-70 in the 0.5 mM, 5 mM, and 50 mM buffers at 1.0 M ionic strength and 23.5 °C. The increase in the apparent pK_a_ value of M-70 that occurs with increasing ionic strength of the buffer can be attributed to a salt effect. Salt effects in polymers are well known and have been reported by other workers in the field [18]. The effect of the NaCl concentration on the water content of polyNIPA was previously reported by Park and Hoffman [23]. At high NaCl concentrations, the distances between alkyl acrylic acid monomer units (responsible for the source of charge in the NIPA copolymer) decrease because of the loss of water from the polymer. Hence, the increase in the “apparent pK_a_” of the polymer can be attributed to a decrease in the distance between pH-functional comonomer units in the three-dimensional polymer network due to a decrease in the volume of the polymer, because of the loss of water from the NIPA copolymer at high NaCl concentrations. The fact that significant swelling is observed at high NaCl concentrations indicates that suitably functionalized NIPA copolymers are well-suited for optical pH sensing in ocean and sea water.

### 2.2. Alkyl Acrylic Acids

Although it is only a minor component in the formulation of M-70, MAA can impart the desired functionality to the microgel. Other pH-sensitive functional comonomers often reported in the literature for NIPA are also acrylates, wherein the terminal carboxylic acid is deprotonated as the pH of the solution in contact with the microgel particles is increased. A series of alkyl acrylic acids was investigated at 23.5 °C using the 50 mM buffer at 0.1 M ionic strength to better understand how incorporating pH-sensitive functional comonomers of increasing hydrophobicity can impact the pK_a_ value of these NIPA copolymers. Alkyl acrylic acid comonomers selected for this part of the study include acrylic acid (AA), methacrylic acid (MAA), ethacrylic acid (EAA), and propacrylic acid (PAA). Crosslinked NIPA copolymers were prepared using 10% AA, 10% MAA, 10% EAA, and 10% PAA (see Table 6).

Figure 7 shows pH response profiles (ascending, i.e., swelling) of four NIPA-based polymers prepared by copolymerization of NIPA with AA, MAA, EAA, and PAA. The pH response profiles shift to higher pH as the alkyl chain length of the functional comonomer increases. Figure 8 shows the ascending and descending pH profiles of each pH-sensitive copolymer in this series. Swelling is reversible in the 50 mM buffer at 23.5 °C and 0.1 M ionic strength for M-79 and M-70, whereas it is irreversible for M-80 and M-74. For an irreversible system, our previous experience is that the pK_a_ of the reverse titration (descending pH) is smaller than the pK_a_ of the forward titration (ascending pH), which may be due to a kinetic effect. For M-79, swelling occurs over a larger pH range than for the other copolymers. One possible explanation for this is that fewer acrylic acids are inside M-79 compared to M-70, M-80, and M-74. Therefore, it would take a larger pH change to cause complete swelling. In a previous study, Perera [24] demonstrated using the quartz crystal microbalance (QCM) that g-globulin and hemoglobin could bind to M-79 but not to M-70, M-80, and M-74. Each copolymer was spin-coated onto a QCM gold substrate. After spin coating, the particles were exposed to buffer, and the resonance frequency of the QCM was recorded after 25 min. The QCM substrate was then exposed to a buffer solution containing hemoglobin or g-Globulin. The pH of the buffer used was higher than the pK_a_ of the polymer and less than the pI of the protein. The frequency of the QCM crystal was recorded after 25 min for short-time exposure and after 24 h for long-time exposure. Differences between the frequency of the crystal in contact with buffer solutions with and without hemoglobin or g-globulin are indicative of protein binding, and this only occurs with M-79.

Figure 9 shows kinetics data for the pH-induced shrinking of M-79 and M-70 at 23.5 °C and 0.1 M ionic strength in 50 mM, 5 mM, and 0.5 mM buffers. For M-79, the rate of shrinking is independent of buffer concentration, whereas this rate decreases as the buffer concentration decreases for M-70. This disparity can be explained by the lower amount of AA in the interior of M-79 compared to MAA in M-70 and the fact that pH-induced polymer swelling and shrinking is controlled by the quantity of alkyl acrylic acid in the interior of the NIPA copolymer.

Table 6 summarizes the formulation for each NIPA copolymer, the apparent pK_a_ of each NIPA copolymer, and the octanol-water partition coefficient (which is a measure of hydrophobicity) of each functional comonomer (alkyl acrylic acid). The AA functionalized NIPA copolymer has the lowest apparent pK_a_ of the copolymers investigated in this series and the PAA functionalized copolymer has the largest apparent pK_a_. A shift of 1.9 pH units in the apparent pK_a_ values are observed for the four copolymers in this series. Clearly, the apparent pK_a_ value of these NIPA copolymers can be tuned by increasing the alkyl chain length of the carboxylic acid comonomer.

The apparent pK_a_ of the four NIPA copolymers can be correlated to the Log *p* value (i.e., hydrophobicity) of the pH-sensitive functional comonomer. Less water will be inside the particles containing the more hydrophobic alkyl-acrylic acid, which would explain why the apparent pK_a_ value of the NIPA-propacrylic acid copolymer is the largest in the series. Another possible explanation for the differences in the apparent pK_a_ value of these copolymers is the lower reactivity ratio [25] of EAA and PAA. For all four NIPA copolymers, NIPA is the dominant monomer with only small quantities of the pH-sensitive functional comonomer present. When one monomer is present in greater amounts than another, longer chains containing the abundant monomer are observed with intermittent inclusion of the minor unit. This is true if the relative reactivity ratio of all monomers is the same. For the case of AA and MAA, NIPA is polymerized into chains containing well-separated functional comonomer units as the reactivity ratio of all monomers comprising each of these formulations is approximately the same. However, the pH-sensitive functional comonomers, EAA and PAA, are less reactive as their reactivity ratio is lower. Therefore, the corresponding EAA and PAA copolymers of NIPA would not contain well-separated functional comonomer units. These comonomers would be more closely oriented to each other than to what is observed for linear chains of NIPA and MAA or NIPA and AA. The result is a higher apparent pK_a_ value.

To better understand the thermodynamics of pH-induced swelling and shrinking of the four copolymers in this series, pH response curves were obtained at three different temperatures. The apparent pK_a_ was computed for each response curve (both swelling and shrinking), and the relationship between the apparent pK_a_ (for swelling and shrinking) and temperature was modeled using the Van’t Hoff relationship, see Equation (1), where K_a_ is the apparent acid dissociation constant of the NIPA copolymer, T is the temperature, DH_0_ is the change in enthalpy for the pH-induced swelling of the poly(*N*-isopropylacrylamide) synthate, DS_0_ is the change in entropy associated with pH-induced swelling of the poly(*N*-isopropylacrylamide) synthate, and R is the gas constant.(1)$$\ln K_{a}=\frac{-\Delta H^{0}}{RT}+\frac{\Delta S^{0}}{R}$$

Table 7 and Table 8 list the changes in enthalpy and entropy that occur due to pH-induced swelling and shrinking of the four NIPA copolymers for the 50 mM buffer. The changes in the enthalpy and entropy for the swelling and shrinking of AA and MAA indicate a reversible process, whereas the enthalpy and entropy changes for the swelling and shrinking of EAA and PAA indicate an irreversible process. This is consistent with the ascending and descending pH curves shown in Figure 8 for these four copolymers. The ascending and descending pH curves for AA and MAA overlap, whereas the ascending and descending pH curves for EAA and PAA do not overlap. For PAA, it is likely that M-74 undergoes a conformational change when it swells and does not return to the same initial shrunken state since the turbidity of the M-74 membrane does not return to its original value after the completion of the reverse titration (see Figure 8).

### 2.3. Copolymers of N-Isopropylacrylamide and Propacrylic Acid

Figure 10 shows the pH response profiles (ascending, i.e., swelling) of three copolymers of NIPA and the comonomer PAA: M-103 (2.5%), M-101 (5% PAA), and M-74 (10% PAA), see Table 9. Figure 11 shows the ascending and descending pH profile of each of these copolymers at 23.5 °C and 0.1 M ionic strength using the 50 mM buffer. The range of the pK_a_ values spanned by these three copolymers at 23.5 °C is 5.15 to 5.74 (see Table 9). As the concentration of PAA in the polymer formulation is increased, the apparent pK_a_ increases, and the change in pH occurs over a narrower range as it is not necessary to ionize every alkyl acrylic acid in the polymer to obtain complete swelling at higher PAA concentrations. As for the observed increase in pK_a_ with increasing PAA content, this can be explained by the lower critical solution temperature (LCST) of M-74 compared to M-103 and M-101. The LCST temperature of copolymers of NIPA and PAA [26] decreases with increasing PAA content. pH-sensitive NIPA copolymers above the LCST (e.g., M-74) exist in a collapsed state and will interact less strongly with the buffer than M-101 or M-103 that are below the LCST. Therefore, the apparent pK_a_ of these copolymers can be tuned and their reversibility enhanced (see Figure 11) by adjusting the amount of PAA in the NIPA copolymer formulation.

#### Kinetic Studies of pH-Induced Polymer Swelling and Shrinking

Figure 12 shows rate data for the pH-induced shrinking of M-103 at 23.5 °C and 0.1 M ionic strength in 50 mM, 5 mM, and 0.5 mM buffers.

For M-103, the rate of shrinking is independent of buffer concentration, whereas the rate decreases as the buffer concentration decreases for M-70 (see Figure 9B). This disparity can again be explained by the lower amount of propacrylic acid in the interior of M-103 (2.5% in the formulation of M-103 versus 10% in M-70) and the fact that polymer swelling and shrinking are controlled by the quantity of the alkyl acrylic acid in the interior of the NIPA copolymer.

Kinetics data for the pH-induced swelling of M-70 and M-74 are shown in Figure 13. The pH step used in these kinetic studies (pH 3.3 to pH 6.1 for M-70 and pH 4.5 to pH 7.3 for M-74) corresponds to the regions of the pH profile where the polymer is transitioning between the shrunken state (pH 3.3 for M-70 and pH 4.5 for M-74) and the swollen state (pH 6.1 for M-70 and pH 7.3 for M-74). The different pH values of the buffers used for the M-74 pH step change are due to the higher pK_a_ of M-74. For all three buffer concentrations, swelling (see Figure 13) is faster for M-70 than M-74. M-70 is below the LCST of NIPA, whereas M-74 is at the LCST of NIPA. At 23.5 °C, there are probably areas of the M-74 polymer that have collapsed, and other areas that are swollen. Some propacrylic acids in M-74 would be available to the buffer and others would not be accessible to the buffer. The rate at which the available sites can lose protons depends on buffer concentration, which is consistent with the data shown in Figure 13.

Shrinking (see Figure 14) is also faster for M-70 than M-74 at all three buffer concentrations. M-74 is probably “closed” even at 23.5 °C, so the conjugate acid in the buffer has a harder time penetrating the polymer, which in turn would account for slower protonation. In addition, M-70 may possess some of the attributes of a core–shell particle [27]—a solid core with a pH-sensitive shell that is porous. For this reason, M-70 would be expected to respond faster to changes in the pH of the solution in contact with the membrane than M-74, which consists of only a core that would be denser than the pH-sensitive shell of M-70. More time would be required for M-74 to respond to a change in the pH of the buffer solution in contact with the membrane.

The sigmoid-shaped curve obtained for M-74 with the 0.5 mM buffer (see Figure 14C) suggests two different rates for protonation of the copolymer of propacrylic acid and *N*-isopropylacrylamide. First, the propyl acrylic acids that are most accessible to the 0.5 mM buffer solution are protonated followed by those that are less accessible to the buffer. By comparison, the process is so fast that we cannot differentiate these two rates at 5 mM or 50 mM buffer concentration (see Figure 14A,B).

The rate of polymer swelling and shrinking was also investigated as a function of temperature for M-74. At 37 °C (see Figure 15) all polymer chains of M-74 collapse as M-74 is above the LCST at pH 4.5. The NIPA monomer units interact more strongly with water at 23.5 °C (where the chains are only partially collapsed) than at a higher temperature, 37 °C, so the polymer is more accessible to buffer at lower temperature. Therefore, the slower response of M-74 at higher temperature for swelling, which would appear to be counterintuitive, can be linked to the polymer undergoing a phase change at higher temperature. For polymer shrinking (see Figure 16), the reaction is fast initially at higher temperature but the change in turbidity to the value corresponding to the copolymer in the shrunken state is not observed for any of the buffers over the timescale of the experiment. Although not all propacrylic acids are available for protonation (i.e., shrinking), they become available faster at lower temperatures. As for Figure 16C, the sigmoid-shaped curve observed at lower temperature for the 0.5 mM buffer at 0.1 M ionic strength suggests two different rates for protonation of M-74. The curve recorded at 37 °C in Figure 16C appears to be a superposition of a decaying exponent reaction and an evolving signal with square root time dependence (i.e., diffusion).

## 3. Discussion

Polymer-based sensor technology continues to be an active area of research [28,29]. In this study, the swelling behavior of colloidal polymer particles of *N*-isopropylacrylamide and alkyl acrylic acid has been investigated as a function of the amount of the pH-sensitive functional comonomer in the polymer formulation and the temperature, ionic strength, and concentration of the buffer solution in contact with the particles.

For pH-induced swelling and shrinking, which is the subject of this study, the NIPA polymer chains at low pH possess a compact form that transitions to a configuration consisting of a hydrophobic core surrounded by a more hydrophilic mantle as the pH of the buffer solution in contact with the PVA membrane is increased [30]. At high enough pH, the NIPA polymer chains become extensively uncoiled. pH-induced swelling and shrinking should not be confused with the well-known temperature-induced reversible volume phase transition of NIPA that occurs when the polymer undergoes a change in its organization as the polymer collapses from a coiled structure to a globular one when the temperature exceeds the lower critical solution temperature of the polymer [31].

Previous studies on pH-induced polymer swelling have been restricted to higher buffer concentrations: 50 millimolar and 100 millimolar. The novelty of this study lies in the investigation of pH-sensitive swellable polymers at lower buffer concentrations: 5 millimolar and 0.5 millimolar. Although the sensor response in 5 millimolar buffers is comparable to 50 millimolar and 100 millimolar buffers, the response of the pH-sensitive polymer particles in 0.5 mM exhibits many unique and novel facets.

Swelling and shrinking is observed in a reasonable period for many of the NIPA copolymers investigated in 0.5 mM buffers at 1 M ionic strength, which is truly remarkable. Second, the response rate (i.e., kinetics) of the NIPA-based polymer microspheres to changes in pH decreases as temperature is increased, which is opposite to the behavior that one would expect. Polymer swelling is also irreversible at 1 M ionic strength for all three buffer concentrations investigated (0.5 millimolar, 5 millimolar, and 50 millimolar), whereas it was reversible at 1 M ionic strength in both the 50 millimolar and 100 millimolar buffers investigated in the two previous studies [19,20]. We attribute this to weaker crosslinking of the polymer used in the present study due to the aging of the polymer. This is an intriguing result as increased crosslinking is known to retard swelling, but a certain level of crosslinking is crucial to ensure reversibility as less salt from the 1 M ionic strength buffer will be trapped in the polymer with more efficacious crosslinking. The reduction in the rate of swelling at higher ionic strength is also surprising as the effect of the 1 M buffer is to reduce charge shielding and diminish the activity coefficients of the acid and conjugate base of the buffer. One would expect an increase in the rate of swelling based on this consideration. Finally, the rate and pH range of swelling for the sensor is controlled by the quantity of alkyl acrylic acid in the interior of the NIPA copolymer. By decreasing the alkyl acrylic acid content in the polymer formulation, and hence, in the interior of the polymer, the pH response range of the polymer is increased, which is the opposite to what one would expect. However, we are not asserting that the PNIPA domain is fully accessible to protons and remains “neutral” across the entire pH range.

The apparent pK_a_ of the polymer particles is also dependent upon the amount of alkyl acrylic acid in the copolymer as well as the lower critical solution temperature of the polymer. As in the case of the rate of response to pH, the apparent K_a_ of the polymer particles decreases, not increases, with temperature, which is opposite to what one would expect for the behavior of an organic acid in a medium.

In a previous study [20], polymer particles like M-70 were prepared that were reversible at all ionic strengths, and their responses were independent of the ionic strength of the buffer solution in contact with them. Clearly, the pH-sensitive polymer particles can be tuned to have specific properties based on the specific formulation used to prepare them.

For many of the PVA hydrogel membranes that were investigated in this study, the pH response was reversible. When the hydrogel membrane was initially subjected to a specific pH buffer, the turbidity of the membrane was measured. After the membrane had been subjected to several buffers of increasing pH, the turbidity of the membrane was again measured for the specific pH buffer, and in many cases, the same turbidity value for the membrane was again observed. By comparison, for these same pH solutions, it was sometimes necessary to recalibrate the glass membrane pH electrode after it had been exposed to these buffers.

Significant amounts of pH-induced swelling were observed in buffers of high ionic strength (1.0 M) and low buffer concentration (0.5 mM). Future experiments to better understand polymer swelling in low buffer concentrations at high ionic strength and at higher temperature include substituting CaCl_2_ for NaCl to fix the ionic strength of the buffer and fewer polymer particles in the membrane. This would slow down the rate so that we could better understand what is happening. A proposed application of these polymer particles is monitoring the rising acidity levels in the ocean due to water that has become enriched in carbon dioxide, endangering shell-building organisms by reducing the amount of carbonate available in the water for them.

## 4. Materials and Methods

### 4.1. Materials

Acetonitrile and *N*-isopropylacrylamide were purchased from Acros (Hoboken, NJ, USA). Acrylic acid, methacrylic acid, *N*-*tert*-butylacrylamide, 2,2-Dimethoxy-2-phenylacetophenone, polyvinyl alcohol (MW 85,000–146,000, 98–99% hydrolyzed), and glutaric dialdehyde (50 w% solution in water) were obtained from Aldrich (Milwaukee, WI, USA). Although there may be some alkyl acrylic acid present in the NIPA purchased, the amount was small compared to what was added to the formulations used in these studies. Prior to use, glutaric dialdehyde was diluted to 10% with DI water. Ethacrylic acid and propacrylic acid were prepared using a procedure previously reported in the literature [32]. *N*,*N*-methylenebisacrylamide was purchased from BioRad (Hercules, CA, USA). Acetic acid and hydrochloric acid were obtained from Pharmco, and sodium chloroacetate, 2-(*N*-Morpholino)ethanesulfonic acid (MES), and 3-(*N*-Morpholino)propanesulfonic acid (MOPS) were purchased from Sigma-Aldrich (Atlanta, GA, USA). Sodium hydroxide was purchased from Thermo-Fisher (Waltham, MA, USA). Unless otherwise indicated, all reagents and solvents were used as received.

Chloroacetic acid/sodium chloroacetate solutions were prepared and used as buffers in the pH range 3.0–3.8. Acetic acid and sodium acetate buffer solutions were prepared for the pH range 3.9–5.4. MES was used to prepare pH 5.5–pH 7.3 buffers, and MOPS was used to prepare pH 7.4 to 8 buffers. The recipe for each pH buffer solution was obtained from an online buffer calculator [33].

### 4.2. Synthesis of pH-Sensitive Copolymers of N-Isoproplyacrylamide

pH-sensitive NIPA-co-alkyl acrylic acid copolymers were prepared by free-radical-initiated dispersion photopolymerization. The free radical reaction was initiated using the photosensitive initiator DMPA. The reaction was run under a nitrogen atmosphere at room temperature. The polymer formulation used to prepare the *N*-isopropylacrylamide microsphere particles consisted of a transduction monomer (*N*-isopropylacrylamide), a pH-sensitive functional comonomer (acrylic acid, methacrylic acid, ethacrylic acid, or propacrylic acid), a crosslinker (methylene bisacrylamide) and a lower critical solution temperature modifier (*N*-*tert*-butylacrylamide). To prepare *N*-isopropylacrylamide copolymers of alkyl acrylic acids where the pH-sensitive functional comonomer is either acrylic acid or methacrylic acid, *N*-*tert*-butylacrylamide was necessary to prepare the microspheres. For ethacrylic acid or propacrylic acid, *N*-*tert*-butylacrylamide was not required for the preparation of *N*-isopropylacrylamide-based polymer particles. The solvent used for dispersion polymerization was acetonitrile as all components of the formulation are soluble in it. The total amount of the components (*N*-isopropylacrylamide, alkyl acrylic acid, *N*-*tert*-butylacrylamide, and *N*,*N*-methylenebisacrylamide) in the formulation was 20 millimoles. *N*-isopropylacrylamide, *N*,*N*-methylenebisacrylamide, and *N*-*tert*-butylacrylamide were first added to a 500 mL 3-neck round bottom Pyrex flask containing 80–100 mL of acetonitrile. These three components were agitated for 30 min to ensure their complete dissolution. A closed system was employed to prevent oxygen from infiltrating into the reaction mixture. Upon completion of the dissolution, DMPA and the alkyl acrylic acid comonomer were added to the 500 mL 3-neck round bottom Pyrex flask. After addition of the photo-initiator and the functional comonomer to the flask, the mixture was sonicated for 20 min using a Branson 1510 ultrasonicator (Branson Ultrasonics Corp, Danbury, CT, USA), and the flask was purged with nitrogen to remove any oxygen that may have dissolved in the mixture. Next, the flask was placed in a Rayonet UV photolysis chamber (Southern New England Ultraviolet Company, Branford, CT, USA) and exposed to UV-A (315 nm–400 nm) radiation. The free radical photoreaction was allowed to continue for twelve hours.

The turbid polymer suspension obtained after twelve hours was divided into two 40 mL polypropylene centrifuge tubes and centrifuged at 3000 rpm for 10 min. After disposing of the supernatant, the microspheres were resuspended in a 90/10 (*v*/*v*) methanol and glacial acetic acid solvent. In the new solvent, the centrifuge tubes were sonicated for thirty minutes before being centrifuged again. This washing is repeated a minimum of four times to remove monomers that did not react in the flask. Lastly, the particles are washed with methanol three times and resuspended in a limited amount of methanol, placed in glass vials, and stored between 4 °C and 8 °C until use. Scanning electron microscopy was utilized to characterize the *N*-isopropylacrylamide-co-alkyl acrylic acid copolymers. Each image shows distinct, individual microspheres available for suspension in a polyvinyl alcohol membrane. The particles prepared in these syntheses were generally of uniform size and shape. The size of the polymer particles produced by this procedure varied from 0.5 μm to 1 μm. A scanning electron micrograph of a typical polymer particle prepared using this procedure is shown in Figure 17. These four SEM snapshots were taken at a working distance of 8 mm, 4500 magnification (Figure 17a,c,d) or 2000 magnification (Figure 17b) and 20 KV (Figure 17a,c,d) or 3 KV (Figure 17b) using a JEOL 100 CX-II STEM. (JEOL, Peabody, MA, USA) Each sample was mounted on a gold post to reduce the charge effects caused by the incident electron beam.

### 4.3. Preparation of Polyvinyl Alcohol Membranes

Hydrogel membranes for encasing and immobilizing the swellable pH-sensitive polymer microspheres were prepared by mixing the microspheres with an aqueous polyvinyl alcohol solution (10% *w*/*w*) followed by crosslinking using glutaric dialdehyde. A 10% (*w*/*w*) polyvinyl alcohol solution was prepared in deionized water, and 3-chlorophenol was added to prevent the growth of mold during storage. The amount of 3-chlorophenol added to the polyvinyl alcohol stock solution yielded a 50 ppm solution of the phenol. In the preparation of a membrane, 2 g of a mixture of the *N*-isopropylacrylamide copolymer particles (1%, *w*/*w*) and the polyvinyl alcohol solution (10% *w*/*w*) were prepared in a 4 mL glass vial, and the mixture was magnetically stirred overnight to homogeneously disperse the particles in the polyvinyl alcohol solution. A 50 μL volume of 10% aqueous glutaric dialdehyde (which serves as the crosslinker for the polyvinyl alcohol polymer) was placed in the glass vial, and the contents of the vial were allowed to stir for another two hours. The polymerization reaction for polyvinyl alcohol was initiated using 50 μL of 1 M HCl. The contents of the 4 mL glass vial were allowed to continue stirring for two minutes followed by sonication for two minutes. Once the polyvinyl alcohol polymer became viscous, it was ready for casting to form membranes.

Glass slides (7.5 cm × 2.5 cm) and Teflon tape were used to construct the molds from which the membrane was cast. On one slide, Teflon tape was placed along both longer edges, covering approximately 0.4 cm on each side of the slide, leaving a wide channel through the middle of the bottom slide where the membrane was cast. The approximate thickness of the membrane channel was 127 μm. The contents of the 4 mL glass vial were pipetted onto the membrane channel of the bottom glass slide for casting. A second glass slide was placed on top, and the two slides were clamped using portfolio clips to form a mold for membrane preparation. The two clamped microscope slides were placed in deionized water overnight, which caused the membrane to swell. After the overnight soaking, the portfolio clips were removed, and the pH-sensitive membrane was separated from the glass slide using a razor blade. The membrane was washed again in deionized water for one day. Each membrane was checked for uniformity and irregular segments by measuring the turbidity (700 nm) at every 5.0 mm. Different segments from the same membrane generally gave the same results. Each membrane was then deposited in a 10 mL sample vial containing enough deionized water to remain submerged. The vial was stored at 4 °C in a refrigerator for future use. Further details about the preparation of the membranes and the measurements used to assess membrane uniformity can be found elsewhere [34,35].

PVA hydrogel membranes containing M-70 particles and M-74 have undergone as many as 100 swelling and shrinking cycles without loss of functionality. These membranes have been placed in a glass vial containing distilled water that has been stored away in a refrigerator while retaining their operational effectiveness. Membranes stored away for 10 years appear to have retained their full functionality.

### 4.4. Turbidity Measurements

The swelling and shrinking of the hydrogel membranes prepared from polyvinyl alcohol and the pH-sensitive polyNIPA particles were monitored using a Cary 6000 i spectrometer (Varian Inc., Palo Alto, CA, USA). Although absorbance spectra were collected over the wavelength range 350–700 nm for each membrane, the turbidity profiles were constructed from the absorbance data at 700 nm. Measuring turbidity at longer wavelengths ensured that changes in the refractive index of the membrane are the dominant optical effect as the intensity of the light scattered by the polymer particles decreases with increasing wavelength of the incident light. The swellable pH-sensitive polymers used in this study for sensor fabrication have a resolution of 0.06–0.20 pH units [36].

Each polyvinyl alcohol membrane was mounted onto a custom-built sample holder [26,27] made from black plastic to ensure that it did not contribute to any undue incident light reflections causing stray light bias. Furthermore, black plastic is chemically inert to the buffer solutions in contact with the membrane. Due to the light attenuation that occurs from the membrane holder, a second matched holder without a hydrogel membrane was placed in the reference compartment of the Cary 6000 i spectrometer. A camera image of the “home-built” membrane holder can be found in ref. [17]. To mount a polymer test membrane segment onto a sample holder, a membrane segment was placed in a Petri dish containing deionized water. The segment was then floated onto a microscope slide and smoothed to remove any wrinkles. The membrane was then attached to the inset piece of the sample holder by placing the piece on top of the membrane. This allowed the membrane to completely cover the opening of the inset piece and to adhere to its plastic surface. After the membrane was again investigated for wrinkles or damage, the inset piece, which contains the membrane, was snapped onto the base unit of the sample holder. The complete sample holder, which includes both the base unit and inset piece, was placed into a conventional 1 cm path length cell with the remaining space in the cuvette (1.9 mL) filled with deionized water or buffer solution. The cuvette, which was placed in the sample compartment of the Cary 6000 i spectrometer, was fitted to a custom-built flow cell to allow for presentation of fresh buffer solution to the hydrogel membrane. The flow was regulated using a peristaltic pump (Anko Products, Bradenton, FL, USA) at a rate of 1 mL per minute.

## 5. Conclusions

Lightly crosslinked swellable *N*-isopropylacrylamide polymer particles that respond to pH have been prepared by free radical dispersion polymerization using acrylic acid, methacrylic acid, ethacrylic acid, or propacrylic acid as the pH-sensitive functional comonomer. The diameter of the resulting polymer microspheres was approximately 1 μm. These microspheres were cast into hydrogel membranes prepared by mixing the pH-sensitive polyNIPA particles with aqueous polyvinyl alcohol solutions followed by crosslinking using glutaric dialdehyde for use as pH sensors. When the polymer particles are dispersed in a hydrogel, there are large changes in the turbidity of the membrane as the pH of the solution in contact with the membrane varies. The response time of the polymer particles is rapid and can be attributed to the thickness (approximately 127 μm), the porous nature of the immobilization membrane, and the size and shape of the microspheres. The apparent pK_a_ of these polymer particles increases with temperature and the ionic strength of the buffer solution in contact with the membrane. Furthermore, the pK_a_ of these particles can be tuned by varying the alkyl chain length of the pH-sensitive functional comonomer or the amount of pH-sensitive comonomer in the formulation. Many of the pH-sensitive copolymers investigated in this study exhibited reversible swelling behavior. The swelling behavior of these *N*-isopropylacrylamide copolymers has been studied as a function of the temperature, ionic strength, and concentration of the buffer solution in contact with the membrane, as well as the hydrophobicity of the pH-sensitive comonomer used in the polymer formulation.

## Figures and Tables

**Figure 1 molecules-30-01416-f001:**
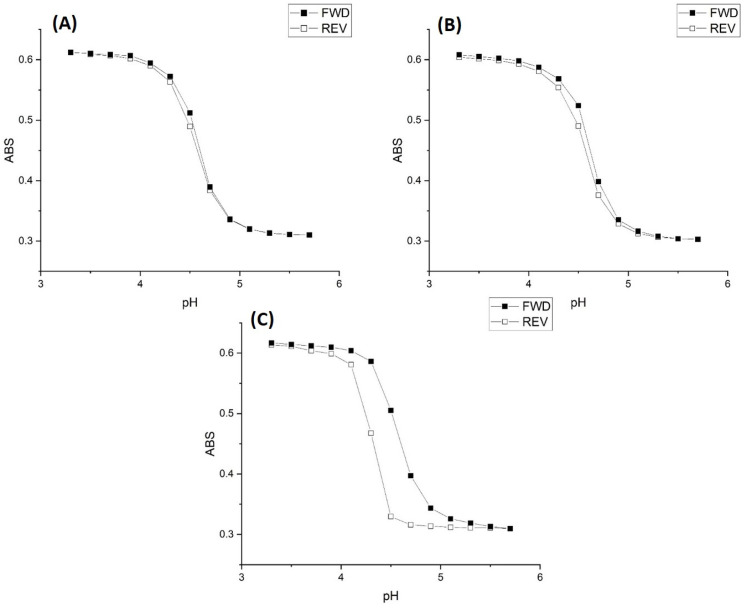
M-70 pH response profile (FWD = ascending and REV = descending) at 23.5 °C and 0.1 M ionic strength for (**A**) 50 mM buffer, (**B**) 5 mM buffer, and (**C**) 0.5 mM buffer.

**Figure 2 molecules-30-01416-f002:**
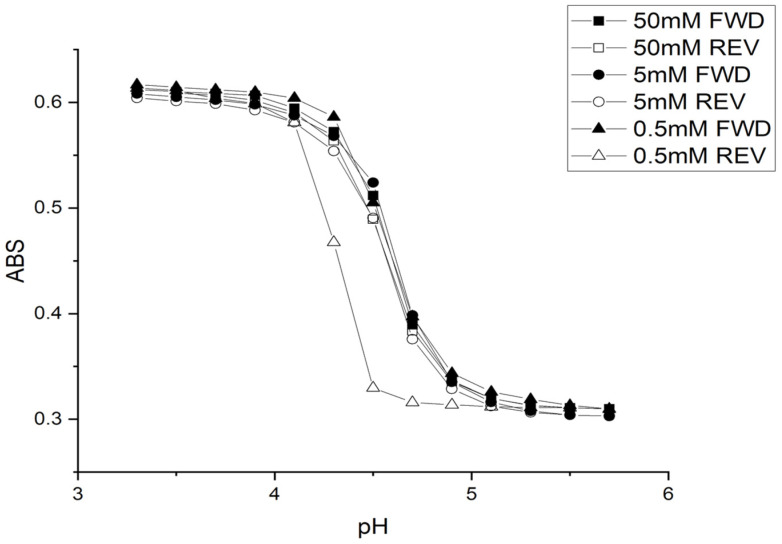
M-70 response profiles (FWD = ascending and REV = descending) at 23.5 °C and 0.1 M ionic strength for 50 millimolar, 5 millimolar, and 0.5 millimolar buffer.

**Figure 3 molecules-30-01416-f003:**
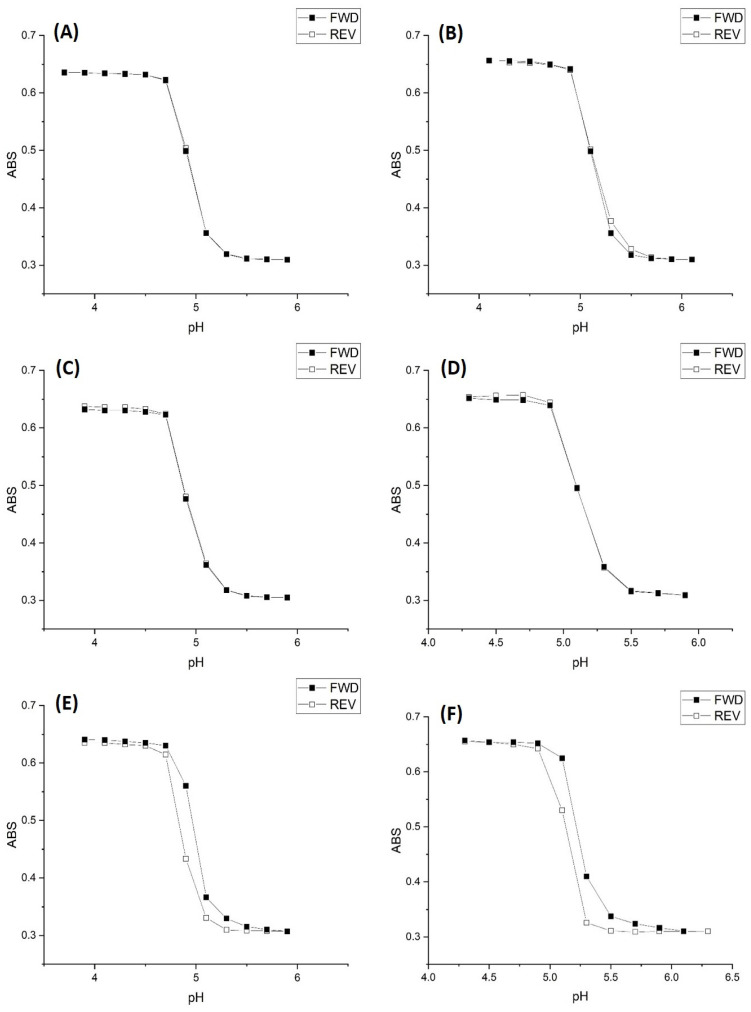
M-70 ascending (FWD) and descending (REV) pH profiles at 0.1 M ionic strength: (**A**) 50 mM buffer at 31 °C, (**B**) 50 mM buffer at 37 °C, (**C**) 5 mM buffer at 31 °C, (**D**) 5 mM buffer at 37 °C, (**E**) 0.5 mM buffer at 31 °C, and (**F**) 0.5 mM buffer at 37 °C.

**Figure 4 molecules-30-01416-f004:**
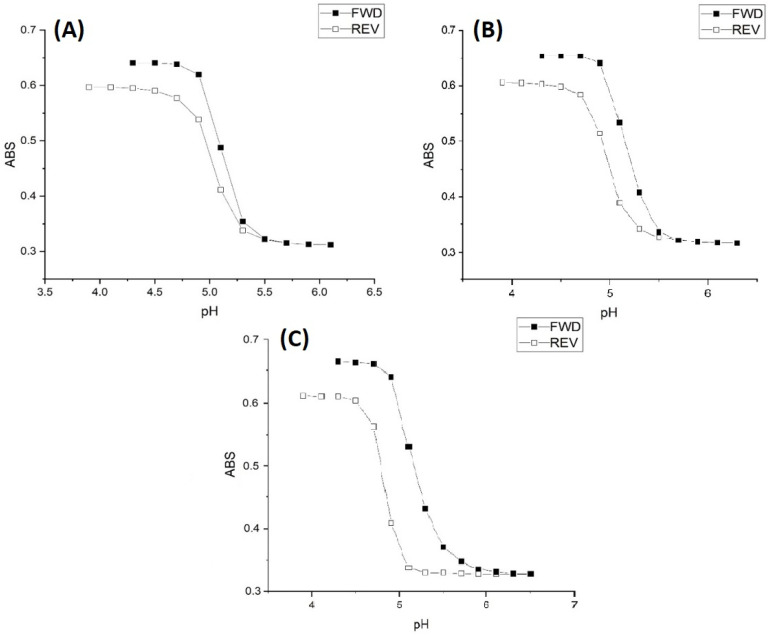
M-70 pH response profiles (both ascending and descending pH) for (**A**) 50 mM, (**B**) 5 mM, and (**C**) 0.5 mM buffers. The ionic strength of each buffer is 1.0 M, and its ionic strength is 23.5 °C.

**Figure 5 molecules-30-01416-f005:**
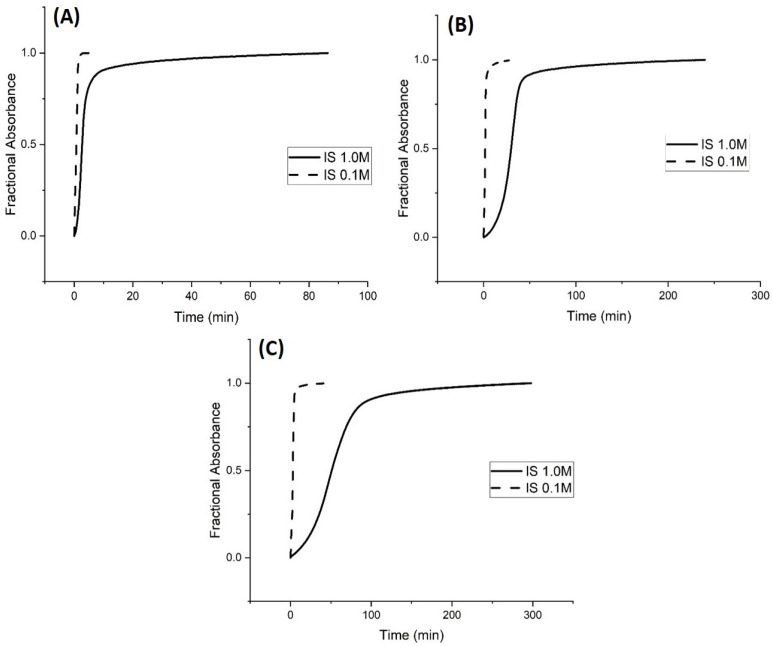
Absorbance versus time at 23.5 °C for (**A**) 50 mM buffer, (**B**) 5 mM buffer, and (**C**) 0.5 mM buffer. pH 3.3 buffer was substituted for pH 6.1 buffer at ionic strengths 1.0 M and 0.1 M to investigate protonation (i.e., the rate of polymer shrinking) for M-70. Ten measurements were collected per second.

**Figure 6 molecules-30-01416-f006:**
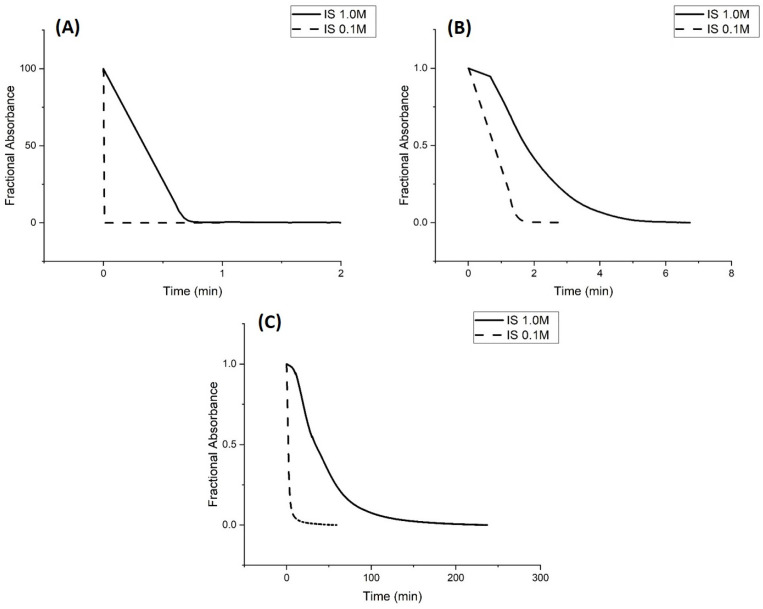
Absorbance (see Equation (1)) versus time for (**A**) 50 mM buffer, (**B**) 5 mM buffer, and (**C**) 0.5 mM buffer at 23.5 °C. pH 6.1 buffer was substituted for pH 3.3 buffer at ionic strengths 1.0 M and 0.1 M to investigate deprotonation (i.e., the rate of polymer swelling) for M-70. Ten measurements were collected per second.

**Figure 7 molecules-30-01416-f007:**
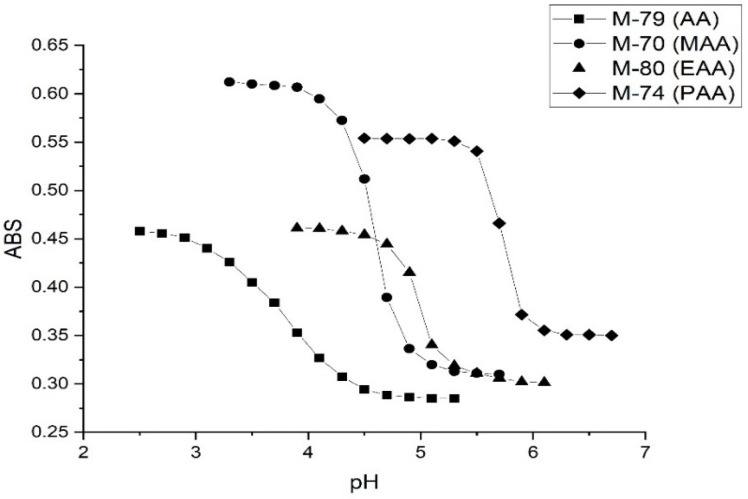
pH response profiles (ascending and swelling) of four poly(*N*-isopropylacrylamides) prepared by copolymerization of NIPA with AA, MAA, EAA, and PAA at 23.5 °C using 50 mM buffer.

**Figure 8 molecules-30-01416-f008:**
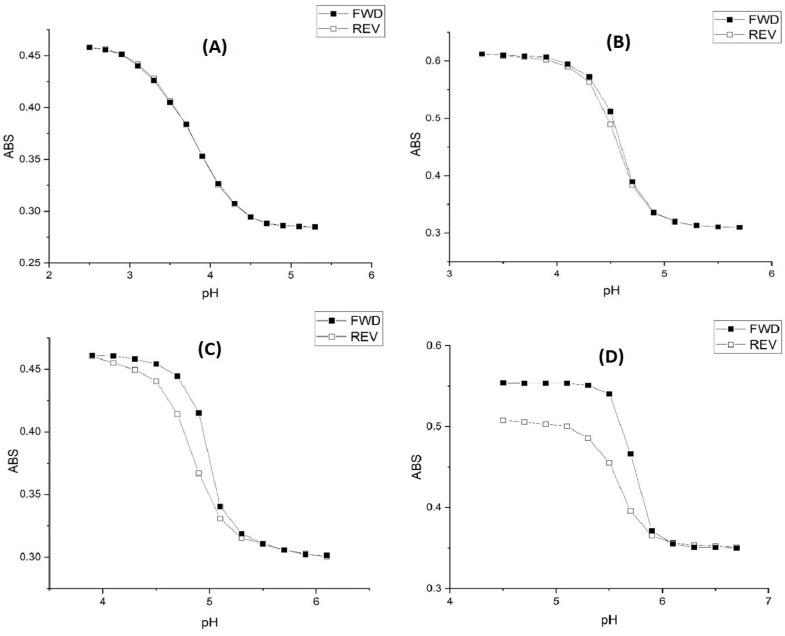
pH response profile (both ascending/swelling and descending/shrinking) of (**A**) M-79, (**B**) M-70, (**C**) M-80, and (**D**) M-84 at 23.5 °C using the 50 mM buffer at 0.1 M ionic strength.

**Figure 9 molecules-30-01416-f009:**
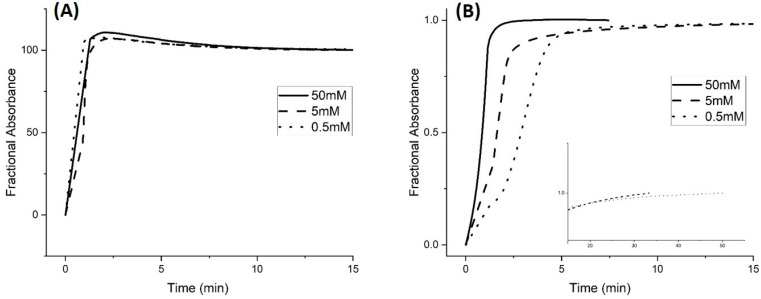
Absorbance versus time for (**A**) M-79 and (**B**) M-70. The temperature was 23.5 °C, the ionic strength was 0.1 M, and the buffer concentration was 50 mM, 5 mM, and 0.5 mM. pH 5.5 buffer was substituted for pH 2.5 buffer in this kinetic experiment. Ten measurements were collected per second.

**Figure 10 molecules-30-01416-f010:**
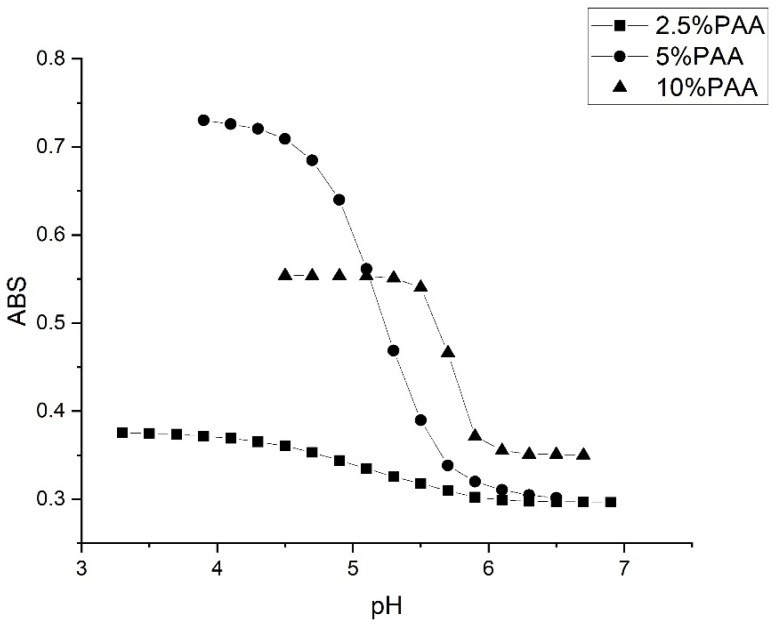
pH response profile (ascending/shrinking) of M-103, M-101, M-74 at 23.5 °C and 0.1 M ionic strength using the 50 mM buffer. Due to less PAA in M-103 and M-101, a larger pH change is required to cause complete swelling.

**Figure 11 molecules-30-01416-f011:**
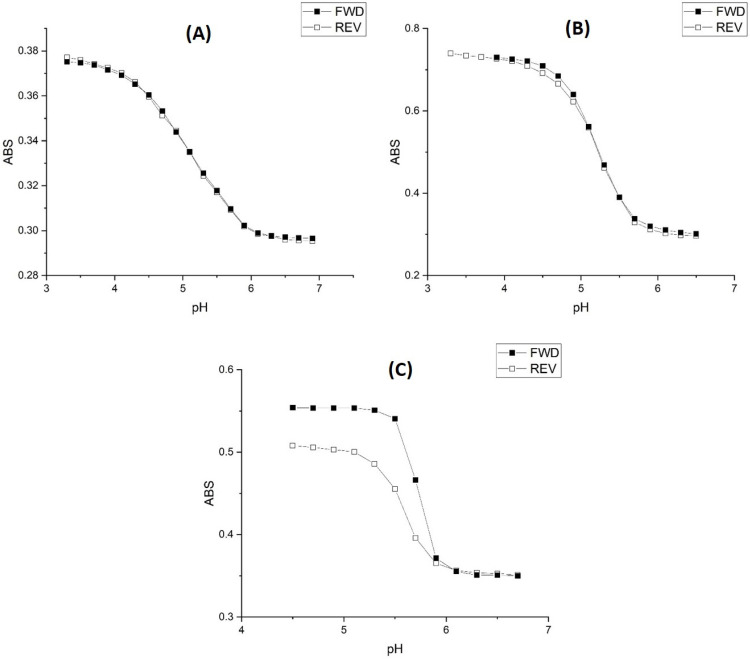
pH response profile (both ascending/swelling and descending/shrinking) at 23.5 °C, 50 mM buffer and 0.1 M ionic strength for (**A**) M-103, (**B**) M-101, and (**C**) M-74.

**Figure 12 molecules-30-01416-f012:**
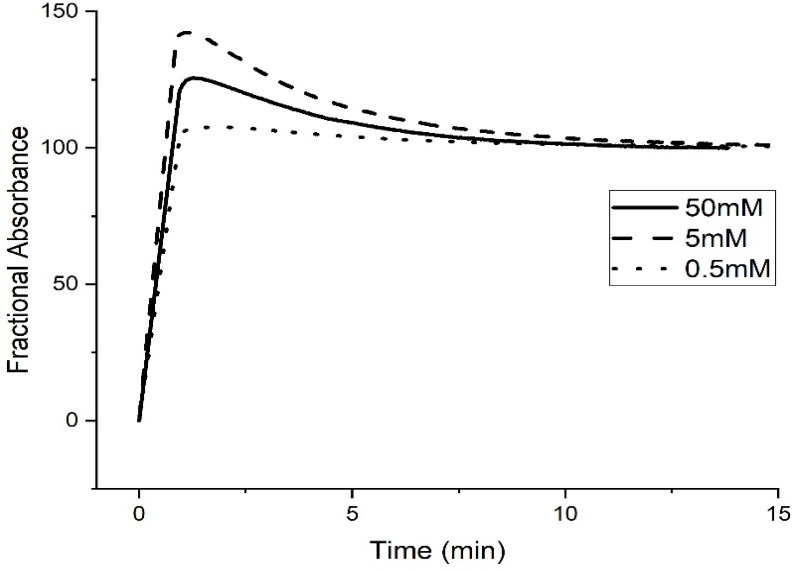
Absorbance versus time for M-103 at 23.5 °C and 0.1 M ionic strength in 50 mM, 5 mM, and 0.5 mM buffers. pH 6.9 buffer was substituted for pH 3.3 buffer. Ten measurements were collected per second.

**Figure 13 molecules-30-01416-f013:**
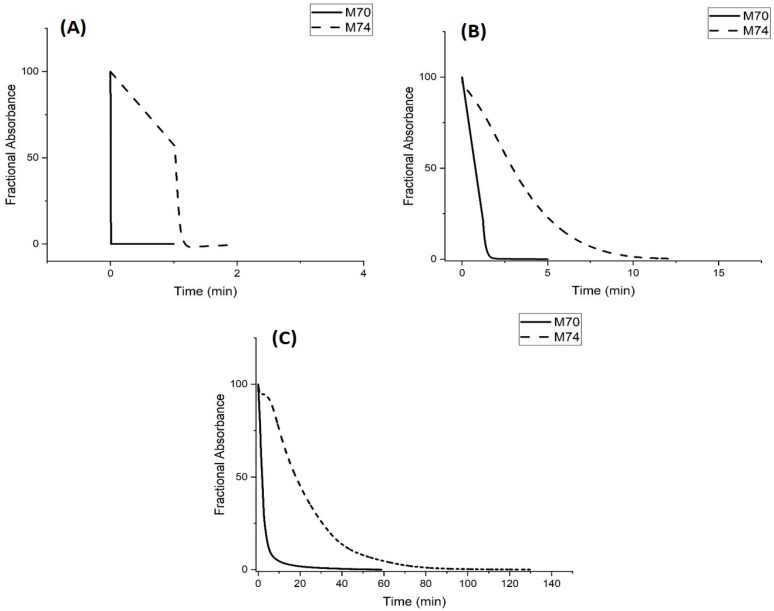
Absorbance versus time for M-70 and M-74. (**A**) 50 mM buffer, (**B**) 5 mM buffer, and (**C**) 0.5 mM buffer. pH 6.1 buffer (0.1 M IS at 23.5 °C) was exchanged for pH 3.3 buffer (0.1 M IS at 23.5 °C) to investigate deprotonation (i.e., the rate of polymer swelling) for M-70, and pH 7.3 buffer (0.1 M IS at 23.5 °C) was exchanged for pH 4.5 buffer (0.1 M IS at 23.5 °C) to investigate deprotonation (i.e., the rate of polymer swelling) for M-74. Ten measurements were collected per second.

**Figure 14 molecules-30-01416-f014:**
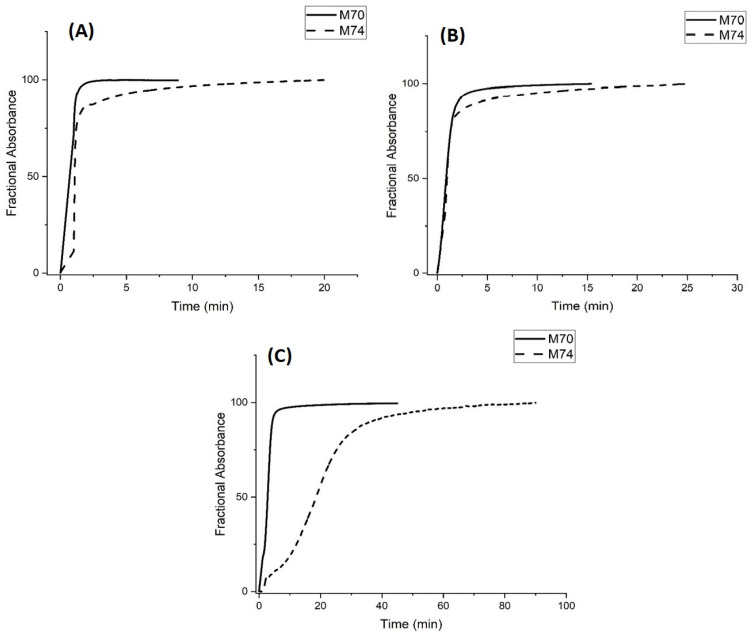
Absorbance versus time for M-70 and M-74. (**A**) 50 mM buffer, (**B**) 5 mM buffer, and (**C**) 0.5 mM buffer. pH 3.3 buffer (0.1 M IS at 23.5 °C) was exchanged for pH 6.1 buffer (0.1 M IS at 23.5 °C) to investigate protonation (i.e., the rate of polymer shrinking) for M-70, and pH 4.5 buffer (0.1 M IS at 23.5 °C) was exchanged for pH 7.3 buffer (0.1 M IS at 23.5 °C) to investigate protonation (i.e., the rate of polymer shrinking) for M-74. Ten measurements were collected per second.

**Figure 15 molecules-30-01416-f015:**
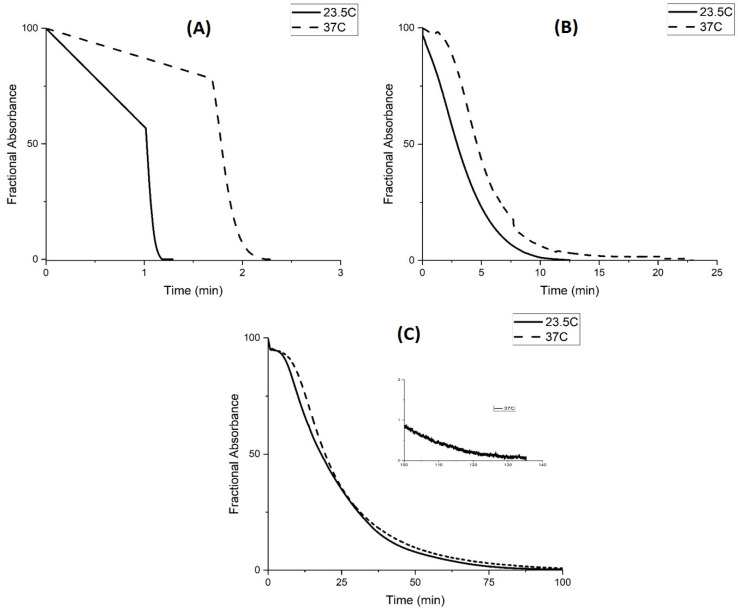
Absorbance versus time for (**A**) 50 mM buffer, (**B**) 5 mM buffer, and (**C**) 0.5 mM buffer. The ionic strength of the buffer was 0.1 M ionic strength at the temperature was 23.5 °C and 37 °C. pH 7.3 buffer was exchanged for pH 4.5 buffer to investigate deprotonation (i.e., the rate of polymer swelling) for M-74. Ten measurements were collected per second.

**Figure 16 molecules-30-01416-f016:**
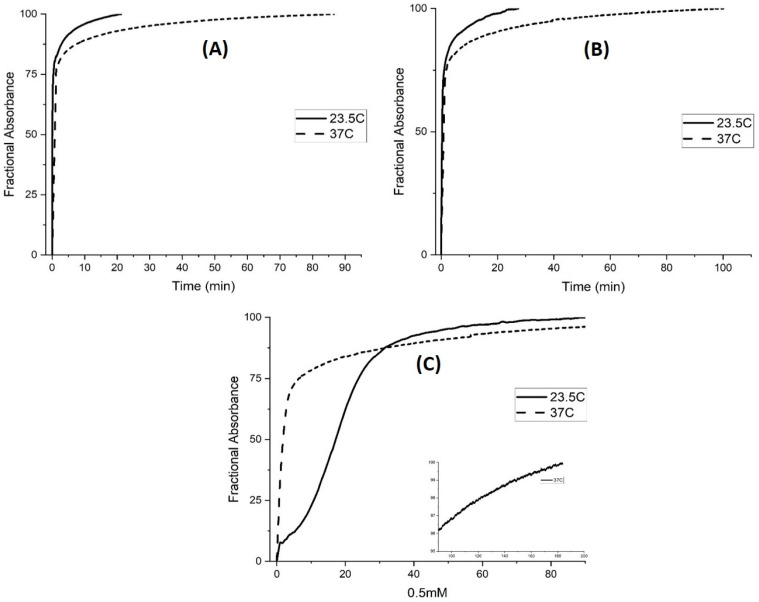
Absorbance versus time for (**A**) 50 mM buffer, (**B**) 5 mM buffer, and (**C**) 0.5 mM buffer. The ionic strength of the buffer was 0.1 M ionic strength at the temperature was 23.5 °C and 37 °C. pH 4.5 buffer was exchanged for pH 7.3 buffer to investigate deprotonation (i.e., the rate of polymer swelling) for M-74. Ten measurements were collected per second.

**Figure 17 molecules-30-01416-f017:**
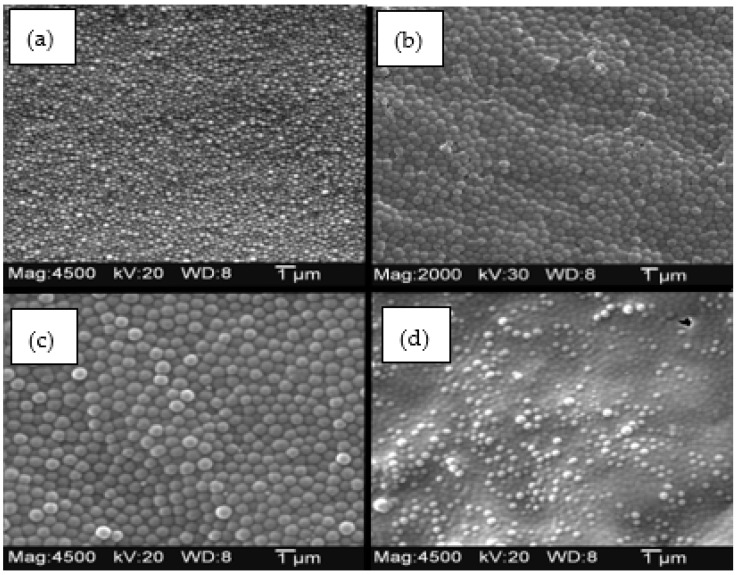
Copolymers of NIPA with (**a**) acrylic acid, (**b**) methacrylic acid, (**c**) ethacrylic acid, (**d**) and propacrylic acid.

**Table 1 molecules-30-01416-t001:** M-70 apparent pK_a_.

Buffer Concentration	Temp	Ionic Strength	FWD	REV
50 mM	23.5 °C	0.1 M	4.59	4.57
5 mM	23.5 °C	0.1 M	4.62	4.59
0.5 mM	23.5 °C	0.1 M	4.57	4.33

**Table 2 molecules-30-01416-t002:** M-70 apparent pK_a_ (50 mM buffer).

Temp	I.S.	FWD (Swelling)	REV (Shrinking)
23.5 °C	0.1 M	4.59	4.57
31 °C	0.1 M	4.93	4.94
37 °C	0.1 M	5.10	5.08

**Table 3 molecules-30-01416-t003:** M-70 apparent pK_a_ (5 mM buffer).

Temp	I.S.	FWD (Swelling)	REV (Shrinking)
23.5 °C	0.1 M	4.62	4.59
31 °C	0.1 M	4.86	4.87
37 °C	0.1 M	5.09	5.09

**Table 4 molecules-30-01416-t004:** M-70 apparent pK_a_ (0.5 mM buffer).

Temp	I.S.	FWD (Swelling)	REV (Shrinking)
23.5 °C	0.1 M	4.57	4.33
31 °C	0.1 M	4.99	4.84
37 °C	0.1 M	5.21	5.17

**Table 5 molecules-30-01416-t005:** M-70 apparent pK_a_ at 1.0 M ionic strength and 23.5 °C.

Buffer Concentration	50 mM	5 mM	0.5 mM
Deprotonation	5.10	5.15	5.15
Protonation	5.02	4.98	4.82

**Table 6 molecules-30-01416-t006:** Formulation of Copolymers of *N*-isopropylacrylamide and Alkyl Acrylic Acid.

NIPA Polymer	Apparent pK_a_ Swelling	Apparent pK_a_ Shrinking	NIPA	Functional Comonomer	NTBA	MBA
M-79	3.84	3.85	14.8 mmoles	2 mmoles of acrylic acid ^1^	2 mmoles	1.2 mmoles
M-70	4.59	4.57	15 mmoles	2 mmoles of methacrylic acid ^2^	2 mmoles	1 mmole
M-80	4.99	4.82	17 mmoles	2 mmoles of ethacrylic acid ^3^	0 mmoles	1 mmole
M-74	5.74	5.60	16.8 mmoles	2 mmoles of propacrylic acid ^4^	0 mmoles	1.2 mmoles

^1^ Log *p* value of acrylic acid (AA) is 0.29; ^2^ Log *p* value of methacrylic acid (MAA) is 0.93; ^3^ Log *p* value of ethacrylic acid (EAA) is 1.08; ^4^ Log *p* value of propacrylic acid (PAA) is 1.59.

**Table 7 molecules-30-01416-t007:** Changes in enthalpy and entropy accompanying swelling.

Polymer	pH Ascending (Swelling) 50 mM Buffer		
M-Series	DH (J/mol)	DS (J/mol)	R^2^
M-79 (AA)	−94,884 ± 12,297	−393.9 ± 40.5	0.9835
M-70 (MAA)	−67,131 ± 7649	−314.5 ± 25.2	0.9872
M-80 (EAA)	−56,297 ± 2452	−284.6 ± 8.1	0.9981
M-74 (PAA)	−59,548 ± 6077	−310.4 ± 20.0	0.9897

**Table 8 molecules-30-01416-t008:** Changes in enthalpy and entropy accompanying shrinking.

Polymer	pH Descending (Shrinking) 50 mM Buffer		
	DH (J/mol)	DS (J/mol)	R^2^
M-79 (AA)	−91,949 ± 8007	−384.0 ± 26.4	0.9925
M-70 (MAA)	−67,472 ± 12,147	−315.4 ± 25.2	0.9686
M-80 (EAA)	−79,096 ± 10,501	−359.4 ± 34.6	0.9827
M-74 (PAA)	−77,740 ± 9853	−369.7 ± 32.4	0.9842

**Table 9 molecules-30-01416-t009:** Composition and apparent pK_a_ of PAA copolymers of NIPA.

Polymer	Apparent pK_a_ Swelling	Apparent pK_a_ Shrinking	NIPA	PAA	MBA
M-103	5.15	5.16	18.5 mmoles	0.5 mmoles	1 mmole
M-101	5.20	5.21	18 mmoles	1 mmole	1 mmole
M-74	5.74	5.60	16.8 mmoles	2 mmoles	1.2 mmoles

## Data Availability

The data presented in this study are available on request from the corresponding author because of University security concerns about housing openly accessible data.

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
