# Peer review of "Thermodynamic and Kinetic Characterization of Colloidal Polymers of N-Isopropylacrylamide and Alkyl Acrylic Acids for Optical pH Sensing"

_molecules, 2025, doi:10.3390/molecules30071416_

Round 1
Reviewer 1 Report
Comments and Suggestions for Authors
Review on MDPI Molecules 3433191
The Authors have prepared and described material for, as they claim, optical sensing of pH. The material (sensing membrane) comprises polymeric microspheres distributed within a hydrogel. The idea is that the material undergoes swelling and shrinking in response to changes of the acidity of the medium (aqueous solution). Dependent on the swelling/shrinking state of the membrane, its turbidity changes, and the respective changes of the absorbance is used as the analytical signal. The preparation and characterization of the material is thoroughly described. Besides optical properties per se, thermodynamic and kinetic data are also obtained. The manuscript is of interest and eventually can be published. However, I recommend a major revision before the manuscript can be re-evaluated.
My major concern is as follows.
The very concept of pH is tricky because it refers to activity of an individual ion, a value which cannot be measured without certain extra-thermodynamic assumptions. Normally, pH is measured with a glass electrode (or a metal oxide electrode, or electrode with a membrane made of plasticized PVC with ionophores). The measured signal is an EMF of a cell with liquid junction. The strength of this approach is that (dependent on the selectivity of the electrode membrane) the measured value does depend on the pH only, i.e. at a given pH the EMF (corrected for the liquid junction potential) does not depend on other components of the solution. Glass pH electrode due to its extremely high selectivity is the best one in this sense. Importantly, although the potentiometric signal of electrodes is governed by ion-exchange processes, the ion exchange within the Nernstian response range is limited to the sensor/sample interface, and the composition of the bulk of the sensor phase (membrane) remains constant over the whole range of the Nernstian response of the electrode.
On the contrary, optical sensors (including those proposed by the Authors) operate in a different way. Ion-exchange process: protonation/deprotonation coupled either with release/sorption of a cation (e.g. Na+), or with sorption/desorption of an anion takes place in the sensor bulk. From thermodynamical and electrochemical point of view, optical sensors are optical analogs of cells without liquid junction, and, indeed, their response is determined not by activity of some individual ion (e.g. H+) but by a ratio of activities of two cations (e.g. H+/Na+) or a multiple of activities of a cation and an anion (e.g. H+·Cl−). The problem is discussed in sufficient detail in http://dx.doi.org/10.1016/j.snb.2014.10.090
Thus, such sensors measure some “conditional pH” or rather the Hammet acidity function of the sample. See also specific comments 5, 7, 9.
This fundamental issue must be thoroughly addressed in the paper, and necessary explanations must be provided, so readers will not be confused.
Besides, I have a number of specific comments listed below.
1. L. 15. “The diameter of the microspheres” – microspheres of the copolymer? “
2. L. 15, 16. “microspheres were cast into hydrogel membranes prepared by mixing the pH sensitive swellable polymer particles” the particles mentioned are the microspheres?
3. L. 64-66. “The microspheres were cast into hydrogel membranes prepared by mixing the pH sensitive swellable polymer particles with aqueous polyvinyl alcohol (PVA) solutions followed by crosslinking the PVA with glutaric dialdehyde for use as pH sensors.” Please, see above.
4. L. 66, 67. “hydrogen ion concentration” Concentration or pH?
5. L. 68-70. “the microspheres swell due to a change in the polymer solvent interaction parameter triggered by the deprotonation of the carboxylic acid group of the alkyl acrylic acid incorporated into the NIPA copolymer” A deprotonation process is obviously coupled with release of some anion or sorption of some cation for the sake of macroscopic electroneutrality.
6. Figures 1, 2, 4, 7, 8, 10, 11C, and especially 3: the density of experimental points in the response domain is too small. The pH should be varied in shorter steps, so the density of points allows to see a curve instead of an abrupt change. From this viewpoint the point distribution in curve for 2.5 PAA in Fig. 10 and Figs. 11A, B appears appropriate.
7. L. 169, 170. “Polymer swelling and shrinking for M-70 is irreversible in all three buffers at 1.0 M ionic strength.” This indicates incomplete equilibrium at high sodium, and indirectly shows Na-dependence of the signal.
8. L. 191-193. “Evidently, more protons are required than can be provided by the 50 mM, 5 mM, or 0.5 mM buffer for protonation of the carboxylate ions as NaCl is added in substantial amounts to the buffer to fix the ionic strength at 1.0 M.” In principle, this may be a result of an interplay between the amount of protons in the solution and the rate of their transfer form the bulk of the solution to the membrane, although diffusion in solution, apparently, is much faster than in the membrane.
9. L. 193-195. “Differences in the rate of pH induced polymer shrinking (see Figure 4) between low and high ionic strength buffer solutions can probably be attributed to charge stabilization of the carboxylate anions by the sodium ions.” Yes, this clearly shows that the measured signal is not only pH, but also Na+ dependent.
10. Fig. 5. The absorbance at IS 0.1 and 1 M should be measured for the same (long, 100 – 300 s) time period to see whether the steady signal is or is not dependent on the IS. The presented data do not allow concluding on this issue.
11. Kinetic curves, often fractured, look puzzling. Any comments?
12. Fig. 16C. Please, describe the inset in the legend.
13. Fig. 16C. Curve recorded at 37 C looks like a superposition of a decaying exponent - reaction and signal evolving with sqrt(time) - diffusion.
Reviewer 2 Report
Comments and Suggestions for Authors
The article explores the synthesis, thermodynamic properties, and kinetic behavior of lightly crosslinked N-isopropylacrylamide (NIPAAm) polymer particles. The research focuses on their application as optical pH sensors, utilizing acrylic, methacrylic, ethacrylic, and propacrylic acids as pH-sensitive co-monomers. The study demonstrates how these polymer particles, incorporated into hydrogel membranes, exhibit rapid and tunable responses to changes in pH, temperature, and ionic strength, making them promising materials for sensing technologies.
The authors successfully synthesized polymer microspheres approximately 1 μm in diameter using free radical dispersion polymerization. These particles were immobilized in hydrogel membranes by mixing with aqueous polyvinyl alcohol solutions and crosslinking with glutaric dialdehyde. The resulting membranes displayed significant changes in turbidity in response to pH variations, indicating effective optical sensing capabilities. The rapid response times observed in buffer solutions of 50 mM and 5 mM concentrations were attributed to the membrane's thickness and porous structure, alongside the size and shape of the polymer microspheres.
A key finding is the tunability of the apparent pKa of these polymers, which increased with temperature and ionic strength. This tunability, along with reversible swelling behavior, underscores the adaptability and robustness of the materials. Furthermore, varying the alkyl chain length or concentration of the pH-sensitive co-monomers enabled further customization of the sensor properties, highlighting the versatility of these polymers.
Despite these strengths, certain limitations in the study should be addressed.
1. Figures 1-3 appear redundant, presenting the same pH response profile data in different representations for various buffer concentrations and temperatures at a constant ionic strength of 0.1 M. Consolidating these into a single representative figure would enhance clarity and streamline the presentation.
2. Additionally, in line 146, the reference to “f) 5 mM buffer at 37 °C” seems questionable—possibly a typographical error meant to indicate a 0.5 mM buffer.
3. Another drawback is the limited involvement of recent publications on the topic. The most recent references in the article are from 2020 and 2021. Inclusion of more recent publications would provide a better understanding of current advances in polymer-based sensor technology and better position the research within the broader scientific community.
In conclusion, the study makes valuable contributions to the development of customizable pH-sensitive polymer sensors with promising optical sensing capabilities. However, addressing the redundancies in figures, clarifying potential data inconsistencies, and incorporating a broader range of literature would significantly enhance the study’s impact and relevance. Further exploration of real-world applications and comparative analyses would also help strengthen the practical applicability of these findings.
Reviewer 3 Report
Comments and Suggestions for Authors
The manuscript “Thermodynamic and Kinetic Characterization of Colloidal Polymers
of N-isopropylacrylamide and Alkyl Acrylic Acids for Optical pH Sensing” describes a new hydrogel with capacity for optical sensing. However, as reported by the authors, they already had published a previous work where the same hydrogel was used and the effects of temperature and ionic strength on the pH response of a series of NIPA copolymers containing different pH-sensitive functional comonomers was investigated at buffer concentrations of 50 millimolar or 100 millimolar. I cannot see novelty from the previous work to this one.
1) Lack of literature to support the statements from line 41 to 55
2) Authors should stress the novelty of this work specially why using 0.5 mM buffer. Since it seems that works worst
3) Figure 1 and 2 should provide in the legend the meaning of FWD and Rev
4) Discussion is very poor. Authors did not compare with previous studies in the literature.
5) Paper order is strange since methods appear between the discussion and conclusion
6) Stability studies should be added
Round 2
Reviewer 1 Report
Comments and Suggestions for Authors
The Authors have thoroughly addressed my specific comments.
However, I do not feel convinced by their answer to my major concern: whether the sensor proposed does measure pH. It is absolutely clear that the sensor does respond to the acidity of the medium, so the pH does affect the sensor response. The problem is whether the response is determined by the pH ONLY.
It can be seen in Fig. 3 in ref. 16 that the curves of the turbidity vs. pH recorded at various ionic strength are close to one another although do not coincide. Variation of the H ion activity and activity coefficient over the chosen ionic strength calculated by the Davies equation (3rd approximation of the Debye-Huckel theory) is not very broad. The log(a) varies from -0.064 at IS 0.05 to -0.030 at IS 1 i.e. in 0.034 only. The activity coefficient varies from 0.863 to 0.934. I wonder whether the scattering between the turbidity responses fits this variation. As to glass pH electrode, it allows measurements of the pH with accuracy better than 0.01 pH.
The claim "For a pH glass electrode, continued recalibration of the electrode would be required as the ionic strength of the buffer solution changes because the response of the glass electrode is governed by an ion exchange process" is not true. The point is that the glass pH electrode does response to the pH only, over a broad pH range, typically from pH 1 to pH 12, some special glasses allow measurements from 0 to 14. Calibration plots obtained in different series of pH buffers, at different ionic strengths coincide within less than 0.5 mV., and no recalibration is needed.
I still believe that the proposed sensor does measure certain function of the sample acidity, but not the pH. It is like litmus paper which does change color, but gives only a rough estimation of the pH because the dependence of the activity coefficients on the ionic strength of the sample is neglected, as well as those of protonated and deprtonated dye in the litmus paper are considered independent on the protonation degree.
I therefore recommend the Authors to address this issue thoroughly and clarify what exactly is measured and what is the accuracy of the pH estimation by their sensor.
Author Response
Comments and Suggestions for Authors
Reviewer 1
The Authors have thoroughly addressed my specific comments. However, I do not feel convinced by their answer to my major concern: whether the sensor proposed does measure pH. It is absolutely clear that the sensor does respond to the acidity of the medium, so the pH does affect the sensor response. The problem is whether the response is determined by the pH ONLY.
I agree with the reviewer that the pH polymers highlighted in this study also respond to large changes in ionic strength when it is fixed in the buffer using NaCl. However, my research group has previously developed polymers that do not respond to changes in the ionic strength of the buffer (see reference 20 in the second revision of the manuscript). Furthermore, the concern raised by the reviewer about the response of the sensor being determined solely by pH is also an issue with the pH glass electrode in low-capacity buffers (0.5mM) containing high concentrations of NaCl, e.g., sea water. These points are made in the paragraphs below.
It can be seen in Fig. 3 in ref. 16 that the curves of the turbidity vs. pH recorded at various ionic strength are close to one another although do not coincide. Variation of the H ion activity and activity coefficient over the chosen ionic strength calculated by the Davies equation (3rd approximation of the Debye-Huckel theory) is not very broad. The log(a) varies from -0.064 at IS 0.05 to -0.030 at IS 1 i.e. in 0.034 only. The activity coefficient varies from 0.863 to 0.934. I wonder whether the scattering between the turbidity responses fits this variation. As to glass pH electrode, it allows measurements of the pH with accuracy better than 0.01 pH.
The small variation that the curves shown in Figure 3 (reference 20 in the second revision of the manuscript) is more likely attributable to cell positioning uncertainty due to small changes in sample positioning (hydrogel membrane) with respect to the beam from the Cary 5000 UV-Visible absorbance spectrometer as the data for each ionic strength was collected at different times over a period of one year (see Javier Galbán, Susana de Marcos, Isabel Sanz Carlos, Ubide Juan Zuriarrain, “Uncertainty in modern spectrophotometers,” Anal. Chem, 2007, 4763-4767)
The claim "For a pH glass electrode, continued recalibration of the electrode would be required as the ionic strength of the buffer solution changes because the response of the glass electrode is governed by an ion exchange process" is not true. The point is that the glass pH electrode does response to the pH only, over a broad pH range, typically from pH 1 to pH 12, some special glasses allow measurements from 0 to 14. Calibration plots obtained in different series of pH buffers, at different ionic strengths coincide within less than 0.5 mV., and no recalibration is needed.
I do not agree with the statement made by the reviewer about the glass electrode responding to pH only. The following statement is an excerpt from the textbook, Galen W Ewing, Instrumental Methods of Chemical Analysis, 4th edition, McGraw Hill, NY. 1975, p. 280-281.
“One shortcoming of the glass pH electrode is that a higher reading is obtained in the presence of large concentrations of alkali metal ions, the sodium error. The previous equations are no longer adequate. The more complicated relation, known as the Nernst-Eisenman equation, is
where the k’s are the selectivity coefficients. An acceptable pH electrode should have a very small k values, 10-3 or less. These coefficients are not strictly constants and cannot be used to calculate corrections for pH values. They are useful in determining the suitability of various electrodes for particular analytical situations.
Eisenman (4) and others have shown that the selectivity coefficients are related to exchange constants K for an ion exchange process taking place at the glass surface and to the diffusion mobilities of the ions within the glass. For example, its considerations is limited to H+ and Na+ ions, the coefficient k1 is given by
where the subscripts denote activities in solution and adsorbed on the glass surface, UNa+ and UH+ are the respective mobilities in the glass.
(4) G. Eisenman, Glass Electrodes for Hydrogen and Other Cations, Dekker, New York, 1967.”
Calibration plots obtained in different series of pH buffers, at different ionic strengths coincide within less than 0.5 mV., and no recalibration is needed.
pH glass membrane electrodes are periodically recalibrated in a laboratory. This is well known and is taught to undergraduate chemistry majors. In my research laboratory, when performing pH measurements using a pH glass membrane electrode, the electrode is standardized using three commercial buffer solutions (typically pH 4, pH 7, and pH 10; each is 50mM.) There is a YouTube video on this subject https://www.youtube.com/watch?v=1TZxS7A6Y1Q.
I still believe that the proposed sensor does measure certain function of the sample acidity, but not the pH. It is like litmus paper which does change color but gives only a rough estimation of the pH because the dependence of the activity coefficients on the ionic strength of the sample is neglected, as well as those of protonated and deprtonated dye in the litmus paper are considered independent on the protonation degree.
I do not agree with the comparison of pH sensitive polymer particles to litmus paper. The pH sensitive polymer can differentiate 0.06 pH units (see R.-I. Stoian, B. K. Lavine, and A. T. Rosenberger, “pH Sensing using Whispering Gallery Modes of a Silica Hollow Bottle Resonator,” Talanta, 2019, 194, 585-590. https://doi.org/10.1016/j.talanta.2018.10.077.
I therefore recommend the Authors to address this issue thoroughly and clarify what exactly is measured and what is the accuracy of the pH estimation by their sensor.
As stated in my previous response, pH induced polymer swelling of NIPA based polymers is controlled by the polymer solvent interaction parameter. This is well documented in literature. By increasing the pH of the buffer in contact with the membrane, additional carboxylic acid groups are deprotonated, changing the polymer solvent interaction parameter and in turn causing the polymer to swell. The degree to which the polymer swells or shrinks is a function of the hydrogen concentration of the buffer solution (i.e., its pH) in contact with the hydrogel membrane.
The reviewer’s previous claim that a pH glass electrode can measure pH to less than 0.01 units based on the Davies equation (3rd approximation of the Debye-Huckel theory) is not consistent with my experience nor the experience of my coauthors using a variety of commercial glass pH electrodes (e.g., Accumet and Orion). Succinctly stated, a buffer whose pH is 4.01 cannot be differentiated from a pH 4.02 buffer using a commercial pH glass electrode and pH meter. My own experience is that I can reliably differentiate pH to within 0.1 units using a commercial pH glass membrane electrode and pH meter. However, the claim made that one can differentiate pH to within 0.01 units using a commercial glass pH electrode and meter is contrary to my own experience as both a researcher and teacher.
The following sentence has been added to the revised manuscript in the Methods Section, “The swellable pH sensitive polymers used in this study for sensor fabrication have a resolution of 0.06 – 0.20 pH units [34].”
In conclusion, the response of the polymers investigated in this study is influenced by the high concentration of sodium ions (at high ionic strength) in the buffer as is the pH glass membrane electrode, which would also suffer from drift under these conditions (0.5 mM buffer concentration and 1 M IS)
Reviewer 3 Report
Comments and Suggestions for Authors
After reading the authors report, I can't see the novelty of the paper comparing with the previous publication. Despite discussion being facultative, if it is included in the paper it stills has to be supported with literature.
Author Response
Comments and Suggestions for Authors
Reviewer 3
After reading the authors report, I can't see the novelty of the paper compared with the previous publication.
The following material has been added to the Discussion Section of this manuscript to highlight the novel results that were obtained in this study.
Previous studies on pH induced polymer swelling have been restricted to higher buffer concentrations: 50 millimolar and 100 millimolar. The novelty of this study lies in the investigation of pH sensitive swellable polymers at lower buffer concentrations: 5 millimolar and 0.5 millimolar. Although the sensor response in 5 millimolar buffers is comparable to 50 millimolar and 100 millimolar buffers, the response of the pH sensitive polymer particles in 0.5mM exhibit many unique and novel facets. Swelling and shrinking is observed in a reasonable period for many of the NIPA copolymers investigated in 0.5 mM buffers at 1M ionic strength which is truly remarkable. Second, the response rate (i.e., kinetics) of the NIPA based polymer microspheres to changes in pH decreases as temperature is increased which is opposite to the behavior that one would expect. Polymer swelling is also irreversible at 1 M ionic strength for all three buffer concentrations investigated (0.5 millimolar, 5 millimolar and 50 millimolar), whereas it was reversible at 1 M ionic strength in both the 50 millimolar and 100 millimolar buffers investigated in the two previous studies [19, 20]. We attribute this to weaker cross linking of the polymer used in the present study due to the aging of the polymer. This is an intriguing result as increased cross linking is known to retard swelling but a certain level of cross linking is crucial to ensure reversibility as less salt from the 1M ionic strength buffer will be trapped in the polymer with more efficacious cross linking. The reduction in the rate of swelling at higher ionic strength is also surprising as the effect of the 1 M buffer is to reduce charge shielding and diminish the activity coefficients of the acid and conjugate base of the buffer. One would expect an increase in the rate of swelling based on this consideration. Finally, the rate and pH range of swelling for the sensor is controlled by the quantity of alkyl acrylic acid in the interior of the NIPA copolymer. By decreasing the alkyl acrylic acid content in the polymer formulation and hence the interior of the polymer, one can increase the pH range of the polymer, which is also the opposite to what one would expect.
Despite discussion being facultative, if it is included in the paper, it still has to be supported with literature.
The following references detailing the advantages of this methodology have been added to the newly revised manuscript as per the request of the reviewer
- W. R. Seitz , M. T.V. Rooney, E. W. Miele, H. Wang, N. Kaval, L. Zhang, S. Doherty, S. Milde, J. Lenda, “Derivatized, swellable polymer microspheres for chemical transduction,” Analytica Chimica Acta, 1999, 400, 55–64.
- M. T. V. Rooney and W. R. Seitz, “An optically sensitive membrane for pH based on swellable polymer microspheres in a hydrogel,” Anal. Commun., 1999, 36, 267–270.
- D. Westover, W. R. Seitz, B. K. Lavine, “Synthesis and evaluation of nitrated poly(4-hydroxystyrene microspheres for pH sensing,” Microchem. J. 2003, 74, 121-129.
- M. Benjamin, Swelling of pH Sensitive N-isopropylacrylamide Polymer Particles, M. S. Thesis, Oklahoma State University, May 2012.